# MiRNA profiles in blood plasma from mother-child duos in human biobanks and the implication of sample quality: Circulating miRNAs as potential early markers of child health

**Lene B. Dypås***, **Kristine B. Gützkow, Ann-Karin Olsen, Nur Duale***

Norwegian Institute of Public Health, Oslo, Norway

* lenebrattsti.dypas@fhi.no (LBD); nur.duale@fhi.no (ND)

## Abstract

**Data Availability Statement:** All relevant data are within the manuscript and its Supporting Information files.

### Background

MicroRNAs (miRNAs) have been linked to several diseases and to regulation of almost every biological process. This together with their stability while freely circulating in blood suggests that they could serve as minimal-invasive biomarkers for a wide range of diseases. Successful miRNA-based biomarker discovery in plasma is dependent on controlling sources of preanalytical variation, such as cellular contamination and hemolysis, as they can be major causes of altered miRNA expression levels. Analysis of plasma quality is therefore a crucial step for the best output when searching for novel miRNA biomarkers.

### Methods

Plasma quality was assessed by three different methods in samples from mother-child duos (maternal and cord blood, $N = 2\times38$), with collection and storage methods comparable to large cohort study biobanks. Total RNA was isolated and the expression profiles of 201 miRNAs was obtained by qPCR to identify differentially expressed miRNAs in cord and maternal plasma samples.

### Results

All three methods for quality assurance indicate that the plasma samples used in this study are of high quality with very low levels of contamination, suitable for analysis of circulating miRNAs. We identified 19 significantly differentially expressed miRNAs between cord and maternal plasma samples (paired t-tests, FDR<0.05, and fold change>±1.5), and we observed low correlation of miRNA transcript levels between cord and maternal samples throughout our dataset.

**Funding:** This work was supported by The Research Council of Norway, NFR-FREE MEDBIO grant, POEMA - POTENTIAL EARLY DIAGNOSTIC MOLECULAR MARKERS OF ADHD: Analysis of miRNA profiles and DNA methylation status in triads (grant no.: 240763/F20). We would also like to thank the European Union Integrated Project NewGeneris, 6th Framework Programme, Priority 5: Food Quality and Safety; Newborns and Genotoxic exposure risks (FOOD-CT-2005-016320). The funders had no role in study design, data collection and analysis, decision to publish, or preparation of the manuscript.

**Competing interests:** The authors have declared that no competing interests exist.

## Conclusions

Our findings suggest that good quality plasma samples suitable for miRNA profiling can be achieved from samples collected and stored by large biobanks. Incorporation of extensive quality control measures, such as those established here, would be beneficial for future projects. The overall low correlation of miRNA expression between cord and maternal samples is an interesting observation, and promising for our future studies on identification of miRNA-based biomarkers in cord blood plasma, considering that these samples were collected at term and some exchange of blood components between cord and maternal blood frequently occur.

## Introduction

The finding of circulating microRNAs (miRNA) in body fluids (blood plasma, serum or urine) [1, 2] has opened up for the possibility of using blood-based miRNAs as a new approach for diagnostic screening. MiRNAs have been linked to cancer and several other diseases [3], and they have been implicated in promoting neurodevelopmental processes, such as neurogenesis, synapse precursor formation, and synaptic plasticity [4, 5]. The stability of miRNAs circulating in blood, and their role as key regulators of almost every biological process, including precise control of neuronal gene expression and thus fine-tuning signaling pathways [6], suggests that they could serve as non-invasive biomarkers for a wide range of diseases [7–9], and possibly help explain basic disease etiology. Furthermore, miRNAs have been shown to be targeted by epigenetic modification, and in turn, miRNAs can target regulators of epigenetic pathways [10, 11].

Blood-based biobanks, which include collection of umbilical cord blood, such as the Mother, Father and Child Cohort Study (MoBa) biobank [12] at the Norwegian Institute of Public Health (NIPH), and other multi-center studies, increasingly incorporate studies identifying candidate mRNA and miRNA expression profile-based biomarkers for a wide range of disorders. The combination of biological specimens and questionnaire data on lifestyle and exposures in MoBa provide unique possibilities to study the effects of many factors of relevance for pregnancy outcome, health and disease. Identification of biomarkers in an easily accessible and minimally invasive biological sample such as umbilical cord blood would be a valuable complementing tool for early diagnostics and prognostics of different types of diseases and other potential adverse health effects. However, reliable quantification of mRNA and miRNA levels requires high-quality RNA, and compromised RNA integrity has been shown to influence such quantification [13–15]. Previously, we reported that intact and high-quality RNA suitable for mRNA profiling analyses was obtained from blood samples collected in Tempus tubes and stored at -80˚C over a period of up to six years [16]. Different RNA isolation methods have also been compared for quality and yield when isolating from blood samples collected in Tempus tubes and stored in biobanks [17]. However, the blood plasma RNA isolation protocol should be optimized in terms of RNA quality and the amount of plasma needed for miRNA expression profile analysis.

In this study we took advantage of plasma samples collected in an earlier pilot study, with routines for sample collection, preparation and storage comparable to those used in MoBa. The aim was to establish a suitable protocol for plasma quality control and isolation of total RNA (including miRNAs) from blood plasma, in order to optimize the use of precious samples collected in MoBa and other biobanks for studies on miRNA expression. When working with plasma it is essential to ensure that the isolated miRNA reflects freely circulating miRNA levels. Poor procedures for blood collection and plasma preparation can cause contamination from

cells, blood platelets, other microparticles (MPs), or from hemolysis, and skew the resulting miRNA profiles [18, 19]. We therefore performed thorough plasma quality assessment using three different methods. In addition, we analyzed miRNA expression levels to investigate differences and similarities in cord and maternal blood plasma, as we are interested in using miRNA expression profiles in cord blood plasma as a reflection of epigenetic conditions in the newborn child.

The knowledge gained from this study will facilitate and optimize plasma quality assurance and total RNA isolation and analysis in future cohort studies on miRNA expression profiles in plasma samples, especially in cord blood. The protocol established in this study will be the basis for a large project where plasma miRNA profiles will be analyzed in 3600 samples to identify early potential markers of ADHD in children and help further elucidate the role of miRNAs in neurodevelopmental disorders, as well as many other complex disease etiologies. A limited literature search in PubMed did not result in any peer-reviewed publications on plasma miRNA profiling analyzed in cord and maternal samples (harvested around the time of birth) at the same time, and to our knowledge, this is the first time such analysis has been performed.

## Methods

### Study population and sample collection

The current study is based on a small pilot cohort study that was established by collecting maternal and cord blood samples from MoBa-enrolled mothers giving birth at the Oslo University Hospital at the ABC-clinic, Ullevål. MoBa is a prospective population-based pregnancy cohort study conducted by the NIPH. Participants were recruited from all over Norway from 1999 to 2008, and the cohort now includes more than 114.500 children, 95.200 mothers, and 75.200 fathers [12]. Biological material in the form of whole blood and plasma has been collected from the mother, the father, and the child (umbilical cord blood) and stored in the biobank, together with extracted DNA, for future use [16, 20, 21].

The pilot cohort study was initiated in order to develop good logistics and establish methods used for a larger sampling project at a later stage [22, 23]. Maternal and cord blood samples (50 pairs) were collected in CPT-tubes (BD Vacutainer® CPT™ Cell Preparation Tube, a blood collection tube containing citrate anticoagulant with FICOLL™) at the hospital immediately after birth and the plasma was pulled off by centrifugation at 1700 *g* for 30 minutes without break. The plasma samples were aliquoted and stored at −20˚C for later analysis. The samples were collected in the period between June 2007 and February 2008. Only 38 of the original 50 sample pairs were used in this study due to sufficient volumes not being available for both maternal and cord samples in some pairs.

The establishment of MoBa and initial data collection was based on a license from the Norwegian Data Protection Agency and approval from the Regional Committee for Medical Research Ethics. The MoBa cohort is currently regulated by the Norwegian Health Registry Act. The pilot study and this current study have been approved by the Regional Committee for Medical Research Ethics and the Data Inspectorate in Norway. Informed consent was collected from pregnant mothers in week 17–19.

### RNA extraction protocol

Total RNA from blood plasma (200 μl) was extracted using miRNeasy Serum/Plasma Kit (QIAGEN, Norway) according to the manufacturer's protocol with some modifications: (i) Thawed plasma samples were centrifuged for 5 minutes at 16,000 *g* at 4˚C before supernatant was transferred to a new tube, (ii) 1.3 ng/μl MS2 carrier RNA was added to the QIAzol Lysis Reagent, and (iii) a custom spike-in control consisting of a pool of three synthetic *C. elegans*

miRNAs (cel-miR-39, -45, and -238) was used. The isolated RNA was stored at– 80˚C until further processing.

## Quantitative real-time PCR assay

cDNA synthesis was performed with 5µl total RNA as template, using miScript II RT Kit (QIAGEN, Norway) according to the manufacturer's protocol. A no reverse transcriptase control (NRT) was included and samples were incubated at 37˚C for 60 min and at 95˚C for 5 min. The cDNA concentration and quality was assessed by a NanoDrop 1000 spectrophotometer (Thermo Fisher, Norway). The prepared cDNA was stored at -20˚C until analysis.

The real-time qPCR analysis was carried out as previously described [17] in 384-well plates using miScript SYBR Green PCR Kit (QIAGEN, Norway) on a CFX384 Touch™ Real-Time PCR Detection System (Bio-Rad, Norway). Serial dilutions of cDNA were prepared to determine the optimal dilution and a 1:40 dilution was selected. All samples, 38 maternal and 38 cord (n = 76), were analyzed on the same 384-well plate to reduce the influence of run-to-run variations. The expression profiles of 204 miRNA assays, including three spike-in controls (cel-miR-39, cel-miR-45, and cel-miR-238), were obtained using this qPCR layout. These miRNAs were selected based on their expression abundance in human blood plasma/serum and literature search [24, 25]. The miRNA-specific primers were designed based on sequences obtained from the miRBase database (miRBase 20 released; http://www.mirbase.org/)) and primers were purchased from Sigma-Aldrich (Sigma-Aldrich, Norway). Non-template controls (NTC) and melting curve (Tm) analysis were included in each qPCR run.

## Plasma quality control

Plasma quality was assessed using three methods: (i) The presence of residual platelets and microparticles in the plasma samples was measured using a CASY Cell Counter and Analyzer Model TT (Roche, Norway). Analysis was performed according to the manufacturer's instructions using a 45 µm capillary, a 1:2000 dilution with CASYton, a sample volume of 200 µl, and two repeated measurements. (ii) The degree of hemolysis was assessed by measuring oxyhemoglobin absorbance at 414 nm. Absorbance (220–750 nm) of the plasma (1.5 µl) was measured on a NanoDrop$^{TM}$ 1000 Spectrophotometer (Fisher Scientific, Norway) using the UV-VIS module. (iii) Additional hemolysis control was assessed by comparing the expression of hsa-miR-451, known to be enriched in erythrocytes, with the expression of hsa-miR-23a, that has been found to be relatively stable in plasma and not affected by hemolysis [24]; i.e $\Delta Cq$ ($Cq$: quantification cycle) = $Cq_{\text{hsa-miR-23a}}$ - $Cq_{\text{hsa-miR-451}}$, as a measure of hemolysis. A $\Delta Cq$ value below five, between five and seven, and above seven, indicates low, moderate, and high possibility of miRNA contamination from hemolysis, respectively.

In the two methods assessing degree of hemolysis, an additional plasma sample was included as reference for hemolysis. Blood was collected from an adult female volunteer and plasma was prepared after vigorously shaking the blood sample to introduce hemolysis.

## Data analysis

The $Cq$-values were recorded with CFX Manager™ Software (Bio-Rad, Norway). The qPCR data analysis was performed as previously described [17] by the comparative $Cq$-method [26, 27] using cel-miR-39, -45, and -238 from the custom spike-in control as reference.

The raw $Cq$-values from 204 miRNAs (including cel-miR-39, -45, and -238) were pre-processed and miRNAs with inadequate measurements, such as miRNAs with contaminated NTCs, multiple melting temperature (T$_m$) peaks, and T$_m$ variation > 1˚C, were removed. $Cq$-values above 37 cycles were considered beyond the limit of detection (LOD) and removed

from downstream analysis. Due to the nature of this study where we are investigating paired data, all miRNAs with missing values for one or more samples were removed to increase strength. The outcome of these strict quality assurance filtering criteria was an expression matrix consisting of 61 miRNAs x 76 samples (38 cord and 38 maternal) that was used in the downstream analysis (S1 Table). The *Cq*-values for target miRNAs were first normalized with the custom spike-in controls, and then normalized relative quantity (NRQ) was calculated by following the formula NRQ = $2^{-\Delta Cq}$ (where $\Delta Cq = Cq$ (target)—*Cq* (global mean), and "global mean" is the mean for each target across all samples).

## Statistical analysis

Statistical analysis of miRNA expression levels was carried out by paired t-tests to compare the expression level in cord samples to maternal samples, followed by multiple testing correction using Benjamini-Hochberg with a false discovery rate (FDR) of 5%. A paired t-test was chosen due to dependency between the mother and child, and due to the objective being to find statistically significant differences within mother-child pairs. Linear regression analysis was also performed, both on the entire data set and for each miRNA, to investigate the degree of correlation for miRNA expression in cord and maternal blood plasma. NRQ-values were $\log_2$ transformed to obtain normal distribution prior to analysis, and normality was evaluated using Normal Quantile plots. Statistical analyses were performed using JMP Pro version 14.1.0 (SAS Institute Inc., Cary, NC, USA) and R version 3.6.1 [28].

## Results and discussion

### Plasma quality control

Successful biomarker discovery depends on controlling sources of preanalytical variation. Plasma quality assessment is therefore important in order to assure that the isolated miRNA reflects freely circulating miRNA, and not contamination from cells or microparticles (MPs) leading to variations in analysis results not related to any biological differences. Suboptimal preanalytical handling of blood plasma can lead to miRNA contamination from cells not thoroughly removed or from high levels of blood platelets and MPs. Hemolysis is another issue that affects blood plasma quality, as contents from erythrocytes then leak out prior to isolation of the plasma. It has been shown that hemolysis may lead to large overestimations of the abundance of miRNAs present in plasma [18]. In addition to this, the concentration of miRNAs in plasma is very low, and combined with the limited availability of biological material from cohort biobanks, it is essential that these small volumes received from a biobank are of good quality.

Plasma quality was assessed using three different approaches; measuring the presence of MPs, assessing the degree of hemolysis by spectrophotometric analysis, and comparing transcript levels of a miRNA abundant in erythrocytes to transcript levels of a miRNA relatively stably expressed in plasma, as a complementary assessment of hemolysis.

First, the presence of residual platelets or MPs in plasma samples from cord and maternal blood was measured using a CASY Cell Counter and Analyzer Model TT. Detected particles were for the most part smaller than 2 μm in diameter, with counts/μl peaking at ~1 μm across samples (Fig 1). This peak is equivalent to blood MPs, which are smaller than 1.5 μm [29]. The source of MPs in blood is plasma membrane fragments, released from cellular components of blood due to cell stimulation, activation, degeneration, and apoptosis. Exosomes, smaller extracellular vesicles (size; 30–150 nm) that carry miRNA, are also present in plasma [30], but their size is below the limit of detection (LOD = 0.7 μm) for our assay. However, we are interested in the total levels of extracellular miRNAs present in plasma, including those encapsulated in exosomes, and no measures were made in the pre-analytical steps to remove these

vesicles. An interesting mention regarding exosomes and miRNAs is a study investigating exosomes in serum from both maternal and cord blood where they found slightly higher concentrations of exosomes in maternal samples compared to cord samples [31].

Blood platelets, on the other hand, are 2–5 μm in size [32], and as shown in Fig 1, blood platelets have been largely removed from the analyzed plasma samples by an extra centrifugation step added to the protocol prior to isolation of RNA. A normal platelet count in blood is between 150.000 and 450.000 platelets/μl, and the observed total particle count in the 2–5 μm size range for the analyzed plasma samples was between 252 and 46.008 counts/μl, with a median of 3.301 counts/μl, suggesting that blood platelets were largely removed from our plasma samples.

Furthermore, maternal samples contained more MPs compared to cord blood plasma (Fig 1) and displayed a large variance between samples. The cord blood samples seem to be more homogenous with respect to MP content. Nine maternal plasma samples (~12% of samples) had particle counts above 10.000 counts/μl, which is considerably higher than the remaining samples. The observed high particle counts in maternal samples is not unexpected, as the number of circulating blood MPs is usually elevated in pregnant women, with a peak in the 3rd trimester, due to enhanced release of MPs stimulated by inflammatory responses [33, 34]. Maternal blood samples used in this study were collected shortly after birth, and it is therefore reasonable to assume that the blood profile is reflective of pregnancy. MP concentration in newborn cord blood has been found comparable with MP concentration in adult (non-pregnant) plasma samples [35].

Secondly, the degree of hemolysis was assessed by measuring distinct oxyhemoglobin absorbance peaks at 414 nm. It has previously been proposed that data on the degree of hemolysis in plasma samples should always be provided when analyzing freely circulating miRNAs [36]. We agree that this is important in the reporting on this field of research, and a simple spectrophotometric analysis can easily be applied to achieve this. Absorbance (220–750 nm) of plasma from

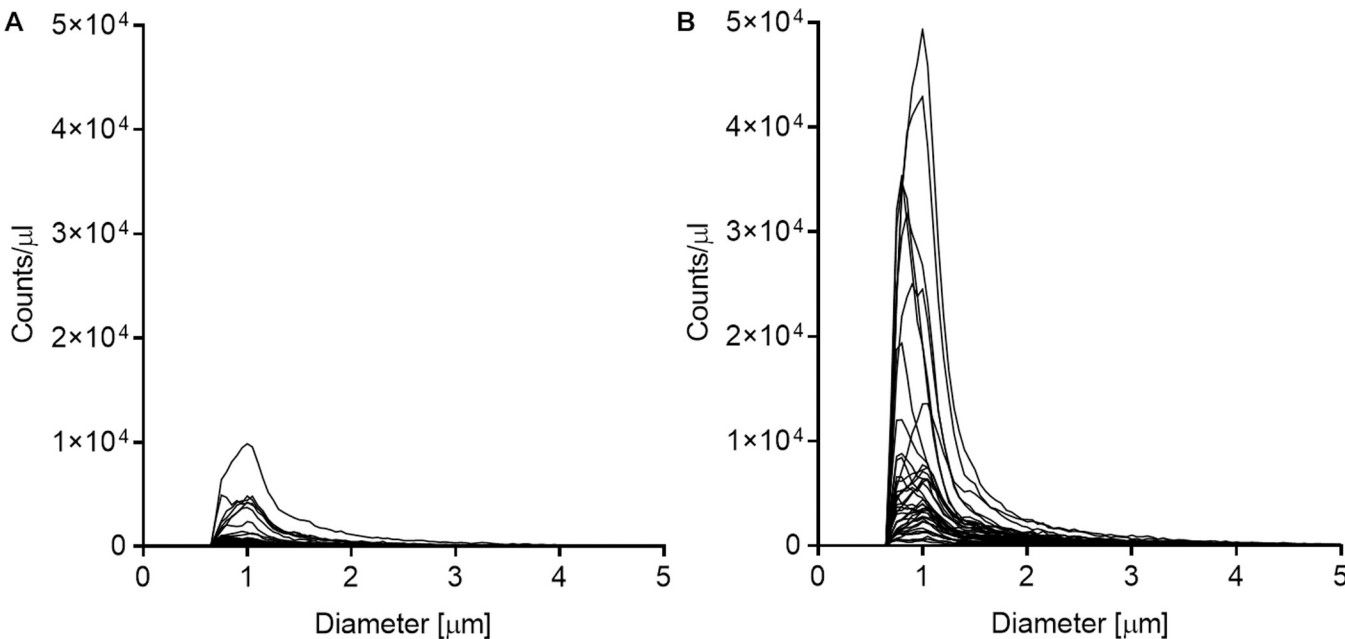

**Fig 1. The presence of residual platelets *and microparticles in plasma sa*mples.** The presence of residual platelets and microparticles in (A) plasma samples from cord (n = 38) and (B) maternal blood (n = 38) was measured using a CASY Cell Counter and Analyzer Model TT with LOD = 0.7. Results are presented as counts/μl by particle diameter (μm).

cord blood, maternal blood, and a reference sample (with intentionally induced hemolysis) was measured on a NanoDrop™ 1000 Spectrophotometer using the UV-VIS module. Absorbance for the hemolyzed reference sample resulted in a distinct peak at 414 nm (Fig 2). When compared with the hemolyzed reference sample, cord and maternal samples had much lower absorbance at 414 nm (Fig 2). The average absorbance at 414 nm for the plasma samples from cord and maternal blood was 0.18 ±0.015 (range: 0.05–0.44) and 0.25 ±0.020 (range: 0.03–0.48), respectively. Absorbance at 414 nm for the hemolyzed reference sample was 1.12.

Thirdly, an alternative and complementary hemolysis assessment was conducted by comparing the transcript level of hsa-miR-451, known to be abundant in erythrocytes, with the transcript level of hsa-miR-23a, which is relatively stably expressed in plasma and unaffected by hemolysis [24]. The calculated $\Delta Cq$ value ($Cq_{\text{hsa-miR-23a}}$ - $Cq_{\text{hsa-miR-451}}$) is a good indicator of the degree of hemolysis. A $\Delta Cq$ value below five, between five and seven, and above seven, indicates low, moderate, and high probability of miRNA contamination from hemolysis, respectively. The average $\Delta Cq$ values for the plasma samples from cord and maternal blood was 2.78 ±0.17 and 2.50 ±0.19, respectively, and all samples had a $\Delta Cq$ below 5 (Fig 3). The $\Delta Cq$-value for the hemolyzed reference sample was 12.13.

Overall, both hemolysis assessment assays suggest that the possibility of contamination by hemolysis is very low and, together with MP concentrations, the results indicate that the plasma samples used in this study are of high quality and suitable for analysis of circulating miRNAs. These are promising results for the MoBa-cohort and other cohorts worldwide, as these samples are collected in hospital settings combined with childbirth, and not in controlled laboratories. We suggest implementation of these plasma QC measures in other blood plasma based biobank studies investigating miRNA expression profiles and biomarker discovery to ensure plasma quality.

## miRNA expression profiling

MiRNAs have been implicated in many diseases and they have been linked to pregnancy-related health outcomes and fetal programming. Their potential in biomarker identification when combined with the vast amount of information available for samples in large biobank

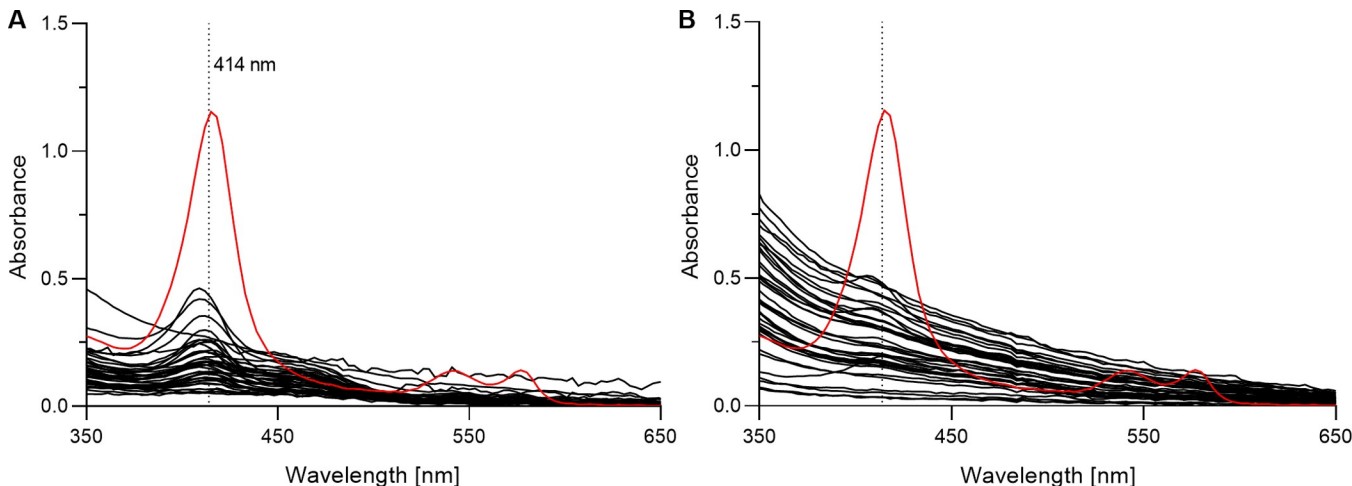

**Fig 2. Spectrophotometric analysis of plasma to assess the degree of hemolysis from oxyhemoglobin absorbance at 414 nm.** Absorbance in plasma was measured for (A) cord blood (n = 38), (B) maternal blood (n = 38), and a hemolyzed reference sample (in red). Average absorbance at 414 nm for the plasma samples from cord and maternal blood was 0.18 ±0.015 (range: 0.05–0.44) and 0.25 ±0.020 (range: 0.03–0.48), respectively. Absorbance at 414 nm for the hemolyzed reference sample was 1.12.

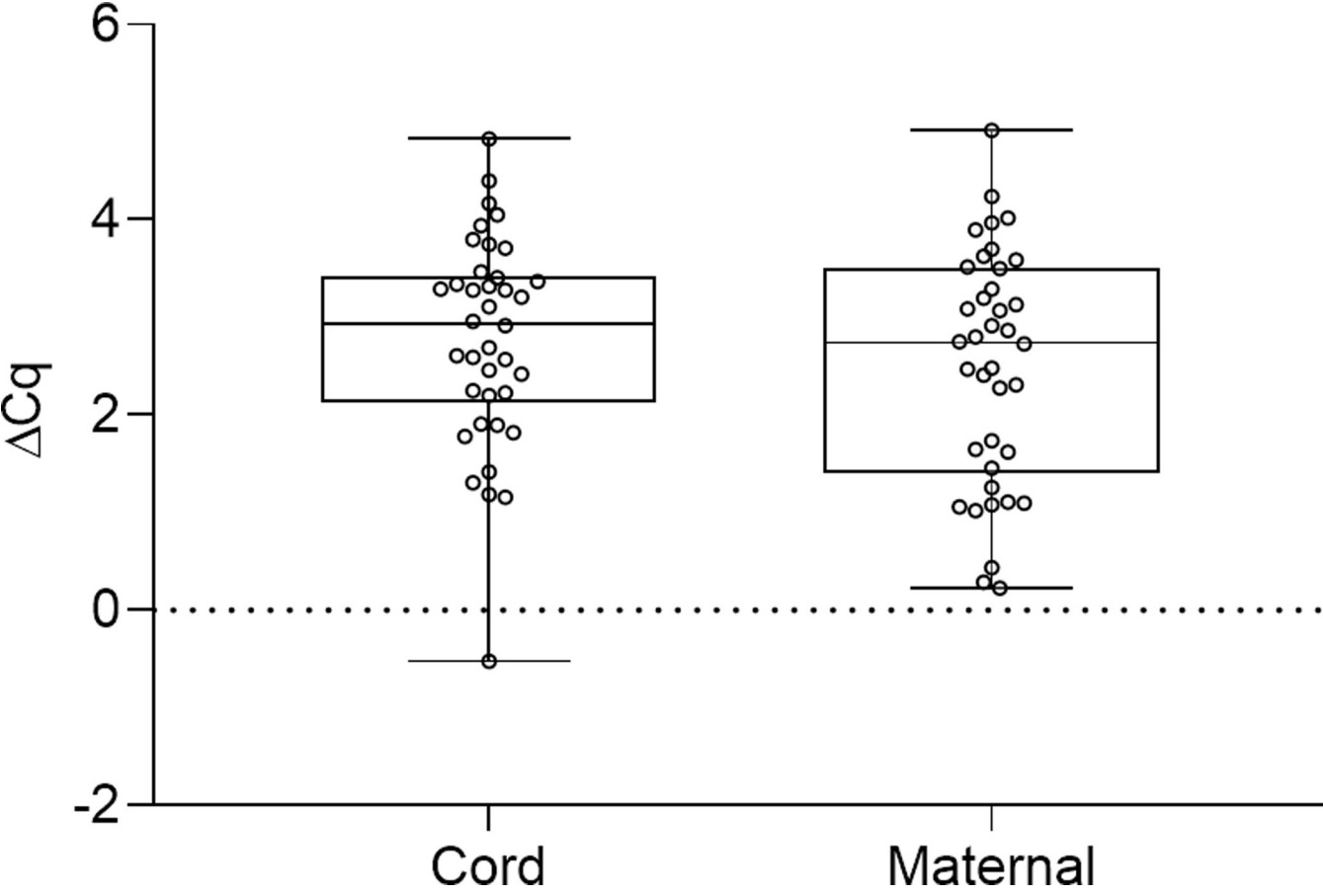

**Fig 3. Distribution of $\Delta Cq = Cq_{\text{hsa-miR-23a}} - Cq_{\text{hsa-miR-451}}$ in plasma samples as a measure of hemolysis.** The transcript level of hsa-miR-23a was compared with the transcript level of hsa-miR-451a to assess the degree of hemolysis in plasma samples from cord blood (n = 38), maternal blood (n = 38), and a hemolyzed reference sample. Average $\Delta Cq$ for the plasma samples from cord and maternal blood was 2.78 ±0.17 and 2.50 ±0.19, respectively. The hemolyzed reference sample had a $\Delta Cq$ of 12.13.

cohort studies provides great opportunities, but these opportunities depend on reliable quantification of miRNA levels. Following an extensive plasma quality assurance, quantification and distribution analysis of circulating miRNAs in plasma from cord and maternal blood in the general population was performed.

The expression profiles of 201 human miRNAs were analyzed by qPCR. After raw data pre-processing and strict quality assurance filtering criteria, an expression matrix consisting of 61 miRNAs x 76 samples (38 cord and 38 maternal) remained and was used in the downstream analysis. Reproducibility is a major area of concern in miRNA research, and to increase strength for our data we removed all miRNAs with missing values, as the nature of this study is to investigate paired data. The quality of the resulting expression matrix is crucial in order to obtain reproducible and high quality data.

In order to compare the miRNA expression levels in cord blood plasma to that of their corresponding maternal sample, paired t-tests were carried out, followed by multiple testing correction using Benjamini-Hochberg with a 5% FDR threshold. Prior to multiple testing correction, the expression levels of 37 miRNAs (61%) were found to be significantly different (p<0.05) in plasma from cord compared to maternal blood samples, and following multiple testing correction this number was reduced to 32 miRNAs (52%). When an additional fold

change cut-off (FC $> \pm$ 1.5) was applied, 19 (31%) of the 32 miRNAs remained (Table 1). One interesting observation was that all these 19 miRNAs had higher transcript levels in cord samples compared to maternal samples

Three samples, two from cord blood and one maternal sample, had consistently high transcript levels across several miRNAs, and may possibly be skewing the results. To confirm this, we generated bar plots and a Tukey box and whiskers plot of median NRQ values per sample, and all plots clearly identify the same three samples to have higher NRQ values when compared to the rest of the sample set (S1 Fig). Median NRQ for the three subjects was 11.45, 11.09, and 10.83, while the median across all subjects was only 1.19. The three identified samples could represent biological extremes, deviating from the general population.

We then performed linear regression analysis to investigate the correlation of expression levels in cord and maternal blood plasma. The observed correlation was low, and increased slightly after removal of the three samples with unusually high transcript levels (as identified above). Linear regression of the $\log_2$-transformed NRQ-values for cord against maternal plasma samples, before and after removal of the three high-expression samples, resulted in an $R^2$ of 0.227 and 0.284, respectively (Fig 4). This lack of strong correlation in the miRNA expression profiles indicates low contamination of cord samples from mother's blood during sample collection. This is an important observation in this study, as one of our future study goals is the utilization of cord blood plasma stored in biobanks to analyze miRNA profiles that may reflect epigenetic conditions in the newborn child. Similarly, low correlation was also observed when linear regression was performed for each miRNA (S2 Table) to find if any single miRNAs might have high correlation in their transcript levels between cord and maternal

**Table 1. miRNAs differentially expressed in cord and maternal blood plasma.**

| miRNA | Mean ±SE | | Fold change | p-value |
|---|---|---|---|---|
| | Cord | Maternal | | |
| hsa-miR-122-5p | 3.54 ±0.60 | 2.29 ±0.60 | 1.54 | 0.0004 |
| hsa-let-7d-3p | 2.15 ±0.32 | 1.38 ±0.24 | 1.55 | 0.0005 |
| hsa-miR-15b-5p | 2.81 ±0.79 | 1.78 ±0.49 | 1.58 | 0.0144 |
| hsa-miR-10b-5p | 2.27 ±0.46 | 1.43 ±0.26 | 1.59 | 0.0089 |
| hsa-miR-423-3p | 3.15 ±0.59 | 1.76 ±0.40 | 1.79 | 0.0006 |
| hsa-miR-423-5p | 3.10 ±0.69 | 1.67 ±0.45 | 1.85 | 0.0007 |
| hsa-miR-222-3p | 2.95 ±0.71 | 1.50 ±0.35 | 1.96 | 0.0220 |
| hsa-miR-335-5p | 4.07 ±1.29 | 2.06 ±0.74 | 1.98 | 0.0013 |
| hsa-miR-151a-5p | 4.07 ±1.47 | 2.01 ±0.76 | 2.02 | 0.0037 |
| hsa-miR-221-3p | 3.80 ±1.04 | 1.75 ±0.46 | 2.17 | 0.0245 |
| hsa-miR-373-5p | 4.91 ±1.91 | 1.94 ±0.67 | 2.52 | 0.0007 |
| hsa-miR-424-5p | 4.85 ±2.21 | 1.92 ±0.37 | 2.52 | 0.0079 |
| hsa-miR-652-3p | 4.70 ±1.59 | 1.73 ±0.50 | 2.72 | 0.0017 |
| hsa-miR-10a-5p | 4.05 ±1.66 | 1.45 ±0.30 | 2.79 | 0.0240 |
| hsa-miR-30e-5p | 3.97 ±1.58 | 1.38 ±0.24 | 2.88 | 0.0080 |
| hsa-miR-365a-3p | 5.60 ±2.98 | 1.42 ±0.25 | 3.95 | 0.0118 |
| hsa-miR-99a-5p | 6.43 ±3.34 | 1.59 ±0.34 | 4.04 | 0.0200 |
| hsa-miR-125a-5p | 5.70 ±3.29 | 1.36 ±0.24 | 4.20 | 0.0209 |
| hsa-miR-125b-5p | 5.81 ±2.76 | 1.35 ±0.23 | 4.29 | 0.0072 |

Mean NRQ, standard error (SE) of the mean, fold change (NRQ$_{cord}$/NRQ$_{maternal}$), and p-value. Paired t-tests followed by multiple testing correction using Benjamini-Hochberg with FDR = 5% and fold change $> \pm 1.5$. Sorted by fold change.

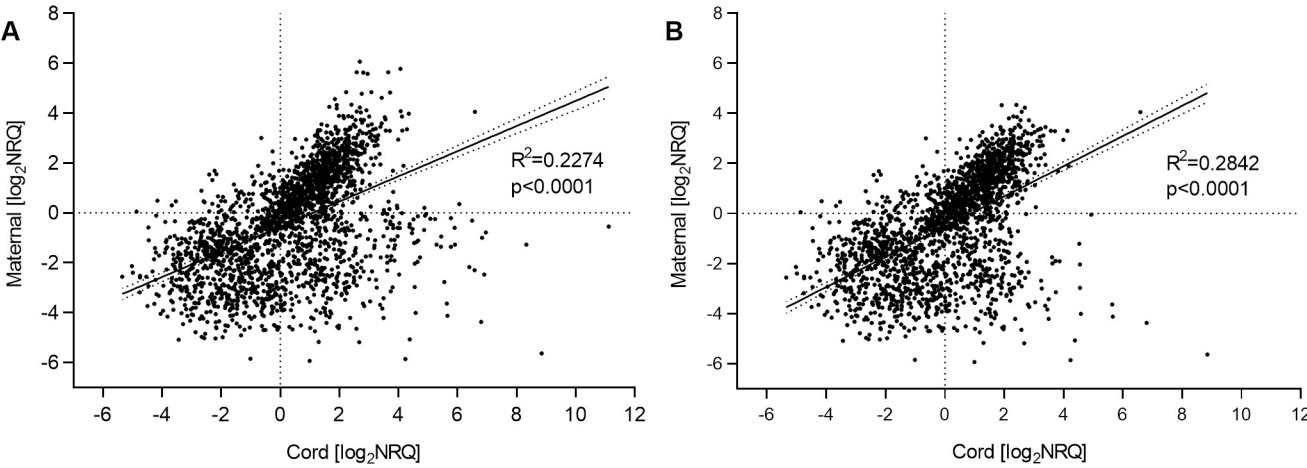

**Fig 4. Correlation of expression levels in plasma from cord and maternal blood.** Linear regression of $\log_2$-transformed NRQ-values for maternal plasma samples as a function of cord samples to assess correlation of expression levels (A) before and (B) after removal of potential outliers.

blood, with hsa-miR-142-3p demonstrating the highest correlation ($R^2$ of 0.637). With low correlation between cord and maternal blood plasma samples, it would be interesting to build a predictive model to discriminate between cord and maternal samples, based on the nineteen-miRNA signature panel identified here. This should be attempted in a later study with a larger sample size, as our data set was too small for both training and validation of a robust predictive model.

Another interesting observation regarding our studies on miRNAs and neurodevelopmental disorders is the miRNAs previously implicated in relation to ADHD. Through a literature search, we identified 30 miRNAs linked to ADHD (S3 Table), and among those 30, thirteen were present in our 61 miRNA expression profile data (Table 2). The presence of these miRNAs in cord blood plasma samples is promising for our future research where we will evaluate miRNA expression profiles in cord plasma as early biomarkers of ADHD, since most previous studies have been focused on whole blood or leukocytes, and on samples from children (6–17 years old) or adults [37–39].

## Concluding remarks

Overall, our findings suggest that good quality plasma samples suitable for miRNA profiling can be achieved from samples collected and stored by large biobanks, such as MoBa, and our extensive, although feasible, quality control measures established here can be incorporated in future projects. When working with large biobanks, the amount of biological material available is often limited, and obtaining high quality blood plasma samples is essential. Especially when studying circulating miRNAs in plasma, where the concentration of these molecules is very low. Employing a stepwise filtering approach removing non-informative miRNAs, we identified 19 miRNAs significantly differentially expressed in cord compared to maternal samples. By investigating cord and maternal samples simultaneously in future studies using a larger sample size, a predictive model to distinguish between cord and maternal samples could possibly be built, perhaps using these 19 miRNAs we identified. Furthermore, the low correlation of miRNA levels between cord and maternal samples holds promise for our future work where we will evaluate the potential of using miRNA expression profiles in cord blood plasma as early biomarkers of ADHD and other neurodevelopmental disorders.

**Table 2. MiRNAs implicated in ADHD and present in our data.**

| miRNA | Mean ±SE | | In ADHD studies | ADHD reference |
|---|---|---|---|---|
| | Cord | Maternal | | |
| hsa-let-7d-3p | 2.15 ±0.32 | 1.38 ±0.24 | Increased in serum, decreased in WB. | [40–42] |
| hsa-let-7d-5p | 2.03 ±0.45 | 1.94 ±0.44 | Increased in serum, decreased in WB. | [40–42] |
| hsa-let-7g-5p | 3.23 ±1.24 | 2.87 ±1.32 | Part of ADHD-prediction model in WBCs | [43] |
| hsa-miR-101-3p | 2.13 ±0.30 | 3.50 ±0.87 | Part of ADHD-prediction model in WBCs, and increased in serum. | [43, 44] |
| hsa-miR-107 | 2.27 ±0.39 | 1.87 ±0.44 | Decreased in WB. | [37] |
| hsa-miR-142-3p | 2.13 ±0.37 | 4.66 ±1.79 | Decreased in plasma when presence of psychiatric disease in the family of ADHD-subject. | [45] |
| hsa-miR-151a-5p | 4.07 ±1.47 | 2.01 ±0.76 | Part of ADHD-prediction model in WBCs. | [43] |
| hsa-miR-18a-5p | 2.63 ±0.53 | 2.86 ±0.89 | Decreased in WB. | [37] |
| hsa-miR-191-5p | 3.03 ±0.91 | 2.81 ±1.43 | Increased in PBMCs. | [38] |
| hsa-miR-22-3p | 2.78 ±0.67 | 2.32 ±0.51 | Decreased in WB. | [37] |
| hsa-miR-24-3p | 4.45 ±2.34 | 1.94 ±0.46 | Decreased in WB. | [37] |
| hsa-miR-26b-5p | 2.95 ±0.78 | 1.67 ±0.41 | Decreased in PBMCs. | [38] |
| hsa-miR-30e-5p | 3.97 ±1.58 | 1.38 ±0.24 | Part of ADHD-prediction model in WBCs, and increased in WBCs. | [39, 43] |

Mean NRQ and standard error (SE) of the mean. WB: Whole blood; WBCs: White blood cells; PBMCs: Peripheral blood mononuclear cells.

## Supporting information

**S1 Fig. Identification of outlier subjects.** Median NRQ values per subject as bar plots for (A) cord and (B) maternal blood, and (C) all samples presented in a Tukey box and whiskers plot. (TIF)

**S1 Table. The 61 miRNAs of the final expression matrix.** Mean and standard error of the mean (SE) for raw $Cq$-values, $Cq$-values normalized with the custom spike-in, and NRQ. P-values from paired t-tests. (DOCX)

**S2 Table. $R^2$ and p-values from linear regression analysis for each miRNA.** (DOCX)

**S3 Table. MiRNAs whose expression levels have been implicated in ADHD in previous research.** WB: Whole blood; WBCs: White blood cells; PBMCs: Peripheral blood mononuclear cells. (DOCX)

## Acknowledgments

The Norwegian Mother, Father and Child Cohort Study is supported by the Norwegian Ministry of Health and Care Services and the Ministry of Education and Research. We are grateful to all the participating families in Norway who take part in this on-going cohort study. We thank Maria Laura Amberger for her assistance in performing the spectrophotometric hemolysis assessment and the microparticle counts.

## Author Contributions

**Conceptualization:** Lene B. Dypås, Nur Duale.

**Data curation:** Lene B. Dypås.

**Formal analysis:** Lene B. Dypås.

**Funding acquisition:** Nur Duale.

**Investigation:** Lene B. Dypås, Nur Duale.

**Methodology:** Lene B. Dypås, Nur Duale.

**Project administration:** Lene B. Dypås, Nur Duale.

**Resources:** Kristine B. Gützkow, Nur Duale.

**Supervision:** Kristine B. Gützkow, Ann-Karin Olsen, Nur Duale.

**Validation:** Lene B. Dypås.

**Visualization:** Lene B. Dypås.

**Writing – original draft:** Lene B. Dypås, Nur Duale.

**Writing – review & editing:** Lene B. Dypås, Kristine B. Gützkow, Ann-Karin Olsen, Nur Duale.

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
