## [Decision Letter · Decision Letter 0]

6 Feb 2020

PONE-D-19-35144

MiRNA profiles in blood plasma from mother-child duos in human biobanks and the implication of sample quality: Circulating miRNAs as potential early markers of child health

PLOS ONE

Dear Miss Dypås,

Thank you for submitting your manuscript to PLOS ONE. After careful consideration, we feel that it has merit but does not fully meet PLOS ONE’s publication criteria as it currently stands. Therefore, we invite you to submit a revised version of the manuscript that addresses the points raised during the review process.

Please address the comments from the two reviewers carefully, and detail where/what changes have been made.

We would appreciate receiving your revised manuscript by Mar 22 2020 11:59PM. To enhance the reproducibility of your results, we recommend that if applicable you deposit your laboratory protocols in protocols.io, where a protocol can be assigned its own identifier (DOI) such that it can be cited independently in the future. For instructions see: http://journals.plos.org/plosone/s/submission-guidelines#loc-laboratory-protocols

We look forward to receiving your revised manuscript.

Kind regards,

Zhong-Cheng Luo

Academic Editor

PLOS ONE

Journal Requirements:

Reviewers' comments:

Reviewer's Responses to Questions

**Comments to the Author**

1. Is the manuscript technically sound, and do the data support the conclusions?

Reviewer #1: Yes

Reviewer #2: Yes

2. Has the statistical analysis been performed appropriately and rigorously? 

Reviewer #1: Yes

Reviewer #2: Yes

3. Have the authors made all data underlying the findings in their manuscript fully available?

Reviewer #1: Yes

Reviewer #2: Yes

4. Is the manuscript presented in an intelligible fashion and written in standard English?

Reviewer #1: Yes

Reviewer #2: Yes

5. Review Comments to the Author

Reviewer #1: Dypas et. al. investigated the quality of a specific cohort of bio-banked plasma samples and the ability to reproducibly isolate miRNA from these samples for downstream quantitative analysis. Utilizing three independent techniques for determining plasma quality, they demonstrate that their plasma samples are of high quality (low MP, low hemolysis) and can be used for quantification of circulating miRNA levels. These experiments were properly controlled, and the manuscript is nicely written. My only major comment is the following:

Although the authors mention performing correlation analysis between each miRNA for the 38 pairs, these data are not included in manuscript. As the data in Figure 4 suggests that there may actually a large group of miRNAs with higher expression levels (>log2(0)) that correlate between cord and maternal samples, it would be important to include the R2 and p-values for each miRNA correlation as a supplement in the manuscript. P-value for the linear regression analysis in Figure 4 should also be included.

Minor comments

(1) In methods, the authors state that they collected 50 pairs of cord blood/maternal blood. However, there are only 38 pairs included in the miRNA analysis, with no explanation for the exclusion of other sample pairs. Please include a statement on why 12 pairs were excluded from miRNA analysis, as this could be an important technical caveat.

(2) No statement of patient consent prior to sample collection. This should be included.

(3) Figure 3: Individual points for each sample should be shown in the box plots.

(4) How might altered levels of exosomes affect miRNA analysis? Is there a difference in exosomes between cord blood and maternal blood?

Reviewer #2: The authors conducted thorough examination of the quality of the collected plasma samples by assessing microparticle counts and hemeolysis; and profiled many miRNAs to identified those expressed in cord and maternal blood plasma. The overall analysis procedure was strict and correct cautions were taken to ensure high quality of the obtained data. The quality of the manuscript is suitable to be published in PLOSOne.

One question:

Line 180 : NRQ = 2-ΔCq (where ΔCq = ΔCq (target) - Cq (global mean)).

Please confirm ΔCq (target) is used but not Cq (target); also “global mean” is not defined.

6. PLOS authors have the option to publish the peer review history of their article (what does this mean?). If published, this will include your full peer review and any attached files.

Reviewer #1: No

Reviewer #2: No

---

## [Author Response · Author response to Decision Letter 0]

9 Mar 2020

Reviewer comments, author responses and manuscript changes

Reviewer #1

Comment 1: Although the authors mention performing correlation analysis between each miRNA for the 38 pairs, these data are not included in manuscript. As the data in Figure 4 suggests that there may actually a large group of miRNAs with higher expression levels (>log2(0)) that correlate between cord and maternal samples, it would be important to include the R2 and p-values for each miRNA correlation as a supplement in the manuscript. P-value for the linear regression analysis in Figure 4 should also be included.

Response: We agree that this is valuable information and Figure 4 has been revised to include the p-value for the linear regression analysis, and, a new table (S2 table) containing the R2 and p-values for each miRNA correlation has been included in supplementary information.

Comment 2: In methods, the authors state that they collected 50 pairs of cord blood/maternal blood. However, there are only 38 pairs included in the miRNA analysis, with no explanation for the exclusion of other sample pairs. Please include a statement on why 12 pairs were excluded from miRNA analysis, as this could be an important technical caveat.

Response: Thank you for pointing out that this important piece of information was missing. The manuscript has been revised to include a statement explaining that 12 pairs were excluded due to insufficient sample volume (line 111-112).

Comment 3: No statement of patient consent prior to sample collection. This should be included.

Response: A statement of patient consent prior to sample collection has been included in the methods section (line 117-118).

Comment 4: Figure 3: Individual points for each sample should be shown in the box plots.

Response: Figure 3 has been revised to include individual points for each sample as we agree that this enhances the readability of the plot and adds value.

Comment 5: How might altered levels of exosomes affect miRNA analysis? Is there a difference in exosomes between cord blood and maternal blood?

Response: The questions raised here is a very interesting topic, with no clear answers as of yet. The manuscript has been revised to include a paragraph on miRNAs in exosomes to clarify (line 224-229).

Early research hypothesized that extracellular miRNAs were protected by encapsulation in exosomes, and that the majority of miRNAs found in plasma or serum stemmed from these vesicles (1). Later it was discovered that 90-99% of extracellular miRNAs were actually associated with proteins in the argonaute family, not exosomes (2, 3). Moreover, it has also been shown that exosomes might actually contain very small amounts of miRNA, contrary to what was hypothesized before (4), and therefore might not affect analysis of total circulating miRNAs to such an extent as previously believed. Either way, when analyzing circulating miRNAs there is no precedence to remove exosomes from plasma, and miRNAs in exosomes are regarded as part of the circulating (i.e. extracellular) miRNAs, and altered levels of exosomes containing miRNAs then reflect changes in the individual’s physical state causing altered levels of circulating miRNAs. However, it is important to always include details of plasma preparation steps in your protocol so that readers know if exosomes are included in the plasma (or other sample material) for reproducibility and comparative purposes.

Regarding your question on a difference in exosomes between cord and maternal blood, not much is known. There is however one small study on this in serum, where they found slightly higher concentrations of exosomes in serum from maternal blood compared to cord blood (5).

Reviewer #2

Comment 1: Line 180 : NRQ = 2-ΔCq (where ΔCq = ΔCq (target) - Cq (global mean)). Please confirm ΔCq (target) is used but not Cq (target); also “global mean” is not defined.

Response: Thank you for pointing this out. ΔCq(target) is a typo, and should be Cq(target) as you correctly assumed. The typo has been corrected and a definition of “global mean” has been added (line 183-184).

References

1. Hunter MP, Ismail N, Zhang X, Aguda BD, Lee EJ, Yu L, et al. Detection of microRNA expression in human peripheral blood microvesicles. PLoS One. 2008;3(11):e3694-e.

2. Arroyo JD, Chevillet JR, Kroh EM, Ruf IK, Pritchard CC, Gibson DF, et al. Argonaute2 complexes carry a population of circulating microRNAs independent of vesicles in human plasma. Proc Natl Acad Sci U S A. 2011;108(12):5003-8.

3. Turchinovich A, Weiz L, Langheinz A, Burwinkel B. Characterization of extracellular circulating microRNA. Nucleic Acids Res. 2011;39(16):7223-33.

4. Chevillet JR, Kang Q, Ruf IK, Briggs HA, Vojtech LN, Hughes SM, et al. Quantitative and stoichiometric analysis of the microRNA content of exosomes. Proc Natl Acad Sci U S A. 2014;111(41):14888-93.

5. Jia L, Zhou X, Huang X, Xu X, Jia Y, Wu Y, et al. Maternal and umbilical cord serum-derived exosomes enhance endothelial cell proliferation and migration. FASEB J. 2018;32(8):4534-43.

---

## [Editor Report · Decision Letter 1]

16 Mar 2020

MiRNA profiles in blood plasma from mother-child duos in human biobanks and the implication of sample quality: Circulating miRNAs as potential early markers of child health

PONE-D-19-35144R1

Dear Dr. Dypås,

We are pleased to inform you that your manuscript has been judged scientifically suitable for publication and will be formally accepted for publication once it complies with all outstanding technical requirements.

With kind regards,

Zhong-Cheng Luo

Academic Editor

PLOS ONE
---

## [Editor Report · Acceptance letter]

20 Mar 2020

PONE-D-19-35144R1 

MiRNA profiles in blood plasma from mother-child duos in human biobanks and the implication of sample quality: Circulating miRNAs as potential early markers of child health 

Dear Dr. Dypås:

I am pleased to inform you that your manuscript has been deemed suitable for publication in PLOS ONE. Congratulations! Your manuscript is now with our production department. 

With kind regards,

on behalf of

Dr. Zhong-Cheng Luo 

Academic Editor

PLOS ONE